# Phenobarbital Induces SLC13A5 Expression through Activation of PXR but Not CAR in Human Primary Hepatocytes

**DOI:** 10.3390/cells10123381

**Published:** 2021-12-01

**Authors:** Zhihui Li, Linhao Li, Scott Heyward, Shuaiqian Men, Meishu Xu, Tatsuya Sueyoshi, Hongbing Wang

**Affiliations:** 1Department of Pharmaceutical Sciences, University of Maryland School of Pharmacy, 20 Penn Street, Baltimore, MD 21201, USA; zli@rx.umaryland.edu (Z.L.); lli@rx.umaryland.edu (L.L.); smen@umaryland.edu (S.M.); 2BioIVT, 1450 S Rolling Road, Halethorpe, MD 21227, USA; SHeyward@BioIVT.com; 3Department of Pharmaceutical Sciences, University of Pittsburgh, Pittsburgh, PA 15261, USA; mex5@pitt.edu; 4Pharmacogenetics Section, Laboratory of Reproductive and Developmental Toxicology, National Institute of Environmental Health Sciences, Research Triangle Park, NC 27709, USA; tatsuya.sueyoshi@nih.gov

**Keywords:** phenobarbital, SLC13A5, PXR, CAR

## Abstract

Phenobarbital (PB), a widely used antiepileptic drug, is known to upregulate the expression of numerous drug-metabolizing enzymes and transporters in the liver primarily via activation of the constitutive androstane receptor (CAR, NR1I3). The solute carrier family 13 member 5 (SLC13A5), a sodium-coupled citrate transporter, plays an important role in intracellular citrate homeostasis that is associated with a number of metabolic syndromes and neurological disorders. Here, we show that PB markedly elevates the expression of SLC13A5 through a pregnane X receptor (PXR)-dependent but CAR-independent signaling pathway. In human primary hepatocytes, the mRNA and protein expression of SLC13A5 was robustly induced by PB treatment, while genetic knockdown or pharmacological inhibition of PXR significantly attenuated this induction. Utilizing genetically modified HepaRG cells, we found that PB induces SLC13A5 expression in both wild type and CAR-knockout HepaRG cells, whereas such induction was fully abolished in the PXR-knockout HepaRG cells. Mechanistically, we identified and functionally characterized three enhancer modules located upstream from the transcription start site or introns of the *SLC13A5* gene that are associated with the regulation of PXR-mediated SLC13A5 induction. Moreover, metformin, a deactivator of PXR, dramatically suppressed PB-mediated induction of hepatic SLC13A5 as well as its activation of the SLC13A5 luciferase reporter activity via PXR. Collectively, these data reveal PB as a potent inducer of SLC13A5 through the activation of PXR but not CAR in human primary hepatocytes.

## 1. Introduction

The solute carrier family 13 member 5 (SLC13A5), also known as Na^+^/citrate cotransporter (NaCT) or mammalian I’m Not Dead Yet (mINDY), is a member of the sodium dicarboxylate/sulfate cotransporter family [1,2], mainly expressed in the liver, followed by the testes, brain, and bone marrows at much lower levels [3,4,5]. Being the primary uptake transporter of citrate, SLC13A5 plays a pivotal role in maintaining the intracellular levels of citrate, a key tricarboxylate intermediate of the tricarboxylic acid (TCA) cycle [6,7] and an important precursor of the biosynthesis of triglycerides, fatty acids, cholesterol, and low density lipoproteins [8,9,10]. Due to the relatively high levels of citrate (100–150 μM) in the human bloodstream [11], the abundance of SLC13A5 on the sinusoidal membrane of hepatocytes determines how efficiently the liver can utilize circulating citrate for energy metabolism, lipid synthesis, and gene regulation. Thus, as an important energy sensor regulating cytosolic citrate levels, the expression and function of SLC13A5 can be changed in response to different metabolic stresses and chemical challenges [12]. Increased expression of SLC13A5 in the liver has been reported in patients with obesity, non-alcoholic fatty liver disease, and type 2 diabetes [13]. On the other hand, low expression of SLC13A5 appears to be beneficial in ameliorating hepatic metabolic disorders and hepatoma cell proliferation [4,14], whilst it is associated with the development of a subgroup of early onset epileptic encephalopathy [15,16].

Recent studies reveal that hepatic expression of SLC13A5 can be induced by endogenous hormones and nutritional factors such as glucagon, interleukin 6 (IL-6), and a high-fat diet as well as xenobiotics such as benzo[a]pyrene (Bap) and rifampicin (RIF) through activation of the cAMP-responsive element-binding protein (CREB), signal transducer and activator of transcription 3 (STAT3), aryl hydrocarbon receptor (AhR), and pregnane X receptor (PXR), respectively [17,18,19]. In an attempt to identify novel nuclear receptors as potential regulators of SLC13A5, we observed a robust induction of SLC13A5 expression in human primary hepatocytes (HPH) by phenobarbital (PB), a prototypical activator of the constitutive androstane receptor (CAR, NR1I3). Subsequent studies reveal, however, that PB-mediated induction of SLC13A5 in the liver is a CAR-independent event.

PB, while used mainly for epilepsy management, has profound effects on the liver and elicits pleiotropic signals to stimulate proliferation of the endoplasmic reticulum, alters cell cycle checkpoint controls, inhibits cell apoptosis, promotes liver cell proliferation in rodents, and affects energy homeostasis [20,21,22,23]. PB has also been widely used as a research tool for inducing hepatic drug metabolism and clearance by coordinating the transcription of genes encoding various drug-metabolizing enzymes, with cytochrome P450 (CYP) 2B genes as its prototypical targets [24,25]. Mechanistically, although PB was characterized as a selective activator of CAR in rodents, it activates both human CAR and PXR and induces the expression of a plethora of hepatic genes associated with drug disposition [26,27]. Notably, while PXR and CAR share many target genes to organize a metabolic defense system in the liver, they also regulate distinctive individual target genes, and have shown differential, even opposite, biological functions in certain aspects of energy metabolism and drug clearance [28,29]. The mechanism underlying PB-mediated induction of SLC13A5 in the human liver is largely unknown.

In the current study, we provide experimental evidence to show that PB treatment markedly induces the expression of SLC13A5 at both mRNA and protein levels in HPH, while 6-(4-chlorophenyl) imidazo[2,1-b] [1,3]thiazole-5-carbaldehyde-*O*-(3,4-dichlorobenzyl)-oxime (CITCO), a selective human CAR agonist, only marginally affects SLC13A5 expression. In HepaRG cells, a surrogate of HPH, PB treatment enhances the expression of SLC13A5 in wild-type (WT) and CAR-knockout (KO) but not in PXR-KO HepaRG cell lines. Utilizing a combination of in silico gene analysis, electrophoretic mobility shift assays, and luciferase reporter experiments, we identify novel PXR response elements in both the promoter and intron regions of the *SLC13A5* gene that contribute to PB-mediated induction. Overall, our results reveal that PB is a robust inducer of hepatic SLC13A5 through activation of the PXR but not CAR signaling.

## 2. Materials and Methods

### 2.1. Chemicals and Biologic Reagents

PB, RIF, CITCO, metformin (Met), GW-4064 (GW), chenodeoxycholic acid (CDCA), troglitazone (TGZ), 17β-estradiol (E2), and sulforaphane (SFN) were bought from Sigma-Aldrich (St. Louis, MO, USA). CINPA1 was obtained from Tocris Bioscience (Minneapolis, MN, USA). 4-((4-(Tert-butyl) phenyl) sulfonyl)-1-(2,5-dimethoxyphenyl)-5-methyl-1*H*-1,2,3-triazole (SPA70) was purchased from AK Scientific, Inc. (Union City, CA, USA). Primers for real-time polymerase chain reaction (RT-PCR) were produced by Integrated DNA Technologies, Inc. (Coralville, IA, USA). The Dual-Luciferase Reporter Assay System was purchased from Promega (Madison, WI, USA). WT, CAR-KO, and PXR-KO HepaRG cell lines, and the specific cell culture medium were obtained from Sigma-Aldrich. All other cell culture reagents were from Life Technologies (Grand Island, NY, USA) or Sigma-Aldrich (St. Louis, MO, USA).

### 2.2. Plasmid Constructions

The pSG5-hPXR expression plasmid was acquired from Dr. Steven Kliewer (University of Texas Southwestern Medical Center, Dallas, TX, USA). The pCR3-hCAR1 + A expression vector and SLC13A5-(DR4-1)_3_ luciferase reporter plasmid were constructed as described previously [19,30]. A fragment containing the distal DR4-1 element, DR4-I1, and DR4-I2 in the introns 6 and 7 of the *SLC13A5* gene were subcloned into the SLC13A5-1kb plasmid, creating a construct containing the *SLC13A5* proximal promoter (1kb), a distal and two intron element, termed SLC13A5-1k+DR4-1\I1\I2. The newly constructed plasmid was sequencing-confirmed. The pRL-TK Renilla luciferase plasmid used to normalize transfection efficiency was from Promega.

### 2.3. HPH and HepaRG Cell Culture and Treatment

HPH were acquired from BioIVT (Baltimore, MD, USA). Hepatocytes with viability over 90% were seeded at the density of 0.75 × 10^6^ cells/well in 12-well collagen-coated plates using seeding medium from BioIVT and cultured at 37 °C in a humidified atmosphere of 5% CO_2_ as described previously [31]. Hepatocytes were treated with solvent (0.1% DMSO), CITCO (1 µM), RIF (10 µM), PB (1 mM), E2 (100 nM), Dex (10 µM), CDCA (100 µM), GW (10 µM), TGZ (10 µM), and SFN (10 µM), or cotreated with SPA70 (2.5 µM), CINPA1 (5 µM), or metformin (0.1, 0.5, 1 mM) for 24 or 48 h before harvesting cells for analysis of RNA or proteins, respectively. WT, CAR-KO, or PXR-KO HepaRG cells were seeded in 12-well plates at 1 × 10^5^ cells/well and cultured for 21 days to induce differentiation according to Sigma-Aldrich’s instructions before initiation of the experiments. Subsequently, a serum-free induction medium containing PB (1 mM) or vehicle control was added to HepaRG cell cultures for another 72 h.

### 2.4. RT-PCR Analysis

Total RNA was extracted from harvested cells using the TRIzol reagent (ThermoFisher, Rockford, IL, USA) and reverse transcribed to generate cDNA using a High Capacity cDNA Archive Kit (Applied Biosystems, Foster City, CA, USA) following the manufacturer’s instructions. RT-PCR analysis was conducted using SYBR Green PCR Mastermix (Qiagen, Germantown, MD, USA) on an ABI StepOnePlus Real-Time PCR System (Applied Biosystems, Foster, CA, USA). Primers for the human SLC13A5, CYP2B6, CYP3A4, PXR, and glyceraldehyde-3-phosphate dehydrogenase (GAPDH) include: SLC13A5, 5′-CTTTGTGGCCACCCTGCTATTC-3′ and 5′-AGCAAATCCGCCCCCTAGT A-3′; CYP2B6, 5′-AGACGCCTTCAATCCTGACC-3′ and 5′-CCTTCACCAAGACAAATCCGC-3′; CYP3A4, 5′-GTGGGGCTTTTATGATGGTCA-3′ and 5′-GCCTCAGATTTCTCACCAACACA-3′; PXR, 5′-AAGCCCAGTGTCAACGCAG-3′ and 5′-GGGTCTTCCGGGTGATCTC-3′; and GAPDH, 5′-CCCATCACCATCTTCCAGGAG-3′ and 5′-GTTGTCATGGATGACCTTGGC-3′. Relative gene expression was calculated based on the following equation: fold over control = 2^∆∆Ct^, in that ∆Ct is the differences of cycle threshold numbers between the target gene and the housekeeping GAPDH, and ∆∆Ct denotes the relative change in these differences between control and treatment groups.

### 2.5. Western Blot Analysis

Total proteins of cell homogenate (20 µg) were loaded on SDS-polyacrylamide gels (4–12%) and then electrophoretically transferred onto polyvinylidene fluoride membranes. Subsequently, membranes were incubated with antibodies against SLC13A5 (diluted 1:200; #ab89181, Abcam Inc., Cambridge, MA, USA), CAR (1:1000, #PP-N4111-00, Perseus Proteomics, Tokyo, Japan), PXR (1:1000, #PP-H4417-00, Perseus Proteomics, Tokyo, Japan), or β-actin (1:5000; #A3854, Sigma-Aldrich, St. Louis, MO, USA). After incubation, membranes were washed and incubated with horseradish peroxidase secondary antibodies (#7076S, Cell Signaling Technology) and were developed with West Pico/Femto chemiluminescent substrate (Thermo-Scientific, Rockford, IL, USA). The complete images of all Western blots were presented as Appendix A.

### 2.6. PXR Knockdown in HPH

The knockdown lentiviral plasmids of PXR (shPXR # TRCN0000021620) and the empty lentiviral vector as a negative control were purchased from Sigma-Aldrich (St. Louis, MO, USA). Cultured HPH were infected with negative control or small hairpin RNA against PXR lentivirus particles that were packaged as described previously elsewhere [19]. Infected hepatocytes were cultured in complete William’s E medium for 48 h and then treated with vehicle control or PB (1 mM) for 24 h. Total RNA was prepared for RT-PCR analysis as described above.

### 2.7. Transfection Assays in Hepatoma Cells and HPH

HepG2 cells in 24-well plates were transfected with a combination of SLC13A5 reporter constructs plus the human PXR or human CAR1 + A expression vector [30] by using X-tremeGENE 9 DNA Transfection Reagent (Roche Diagnostics Corporation, Indianapolis, IN, USA) according to the manufacturer’s instruction. Twenty-four hours post transfection, cells were treated with vehicle control (0.1% DMSO), PB (1mM), CITCO (1 μM), or RIF (10 μM) for another 24 h. Reporter gene activities were assayed for firefly signals normalized against the cotransfected Renilla luciferase using the Dual-Luciferase Kit (Promega, Madison, WI, USA). Separately, HPH plated in 24-well collagen-coated plates were transfected with SLC13A5-1k+DR4-1\I1\I2 or SLC13A5-(DR4-1)_3_ construct in the presence of pRL-TK vector using the Effectene reagent (Qiagen, Germantown, MD, USA) as described elsewhere [32]. Transfected HPH were then treated with vehicle control or PB (1 mM), in the presence and absence of metformin (1 mM) for 24 h. Cell lysates were subjected to dual-luciferase assessment as mentioned above. Reported data are representative of triplicate results acquired from three independent experiments.

### 2.8. Electrophoretic Mobility Shift Assays

Electrophoretic mobility shift assay (EMSA) was conducted as outlined previously [26]. DNA probes of predicted response elements were labeled with [α-^32^P] dATP. PXR and the retinoid x receptor (RXR) proteins were synthesized using the TNT quick-coupled in vitro transcription/translation system (Promega, Madison, WI, USA). The binding reaction was prepared as reported previously by Li et al. [19]. Reaction mixtures were resolved on 5% acrylamide gels and ran in buffer containing 7 mM Tris-HCI (pH 7.5), 3 mM sodium acetate, and 1 mM EDTA at 150 V for 2 h. Thereafter, gels were dried under vacuum and exposed to autoradiography at −70 °C.

### 2.9. Statistical Analysis

All quantitative data are from at least three independent experiments and are expressed as the mean ± S.D. Statistically, comparisons were made using one-way analysis of variance followed by a post hoc Dunnett test or Student’s t test, where appropriate. statistical significance was set at * *p* < 0.05, ** *p* < 0.01, or *** *p* < 0.001.

## 3. Results

### 3.1. PB Induces the Expression of SLC13A5 Gene in HPH

We initially tested the effects of multiple nuclear receptors for their potential regulation of hepatic SLC13A5 expression. HPH were treated with prototypical activators of CAR (PB, CITCO), estrogen receptor (ER; 17β-estradiol), farnesoid X receptor (FXR; CDCA, GW-4064), peroxisome proliferator-activated receptor γ (PPARγ; troglitazone), and the nuclear factor erythroid 2-related factor 2 (Nrf2; sulforaphane). RIF, a human PXR agonist and a known inducer of SLC13A5 [19], was used as a positive control. As shown in Figure 1A, among the activators of selected nuclear receptors, only PB (1 mM) demonstrated robust induction of SLC13A5 mRNA to an extent resembling that by RIF (10 µM). Notably, the selective human CAR activator CITCO only marginally altered the SLC13A5 mRNA level. Subsequent experiments showed that the mRNA level of SLC13A5 was increased by PB in a concentration- and time-dependent manner (Figure 1B,C) that maximized around 1 mM, a concentration typically used for PB-mediated induction of drug-metabolizing enzymes and CAR activation in HPH. The protein levels of SLC13A5 were also elevated upon PB treatment in HPH obtained from multiple liver donors (HPH #166, #173, #175, and #176) (Figure 1D). These results suggest that unlike the selective human CAR activators such as CITCO, PB markedly induces the expression of SLC13A5 in HPH.

### 3.2. PB-Mediated Induction of SLC13A5 Was Repressed by Selective PXR but Not CAR Inhibitors

PB is a dual activator of both CAR and PXR in humans [27]. To investigate the contributions of these two receptors on PB-mediated induction of SLC13A5, human hepatocytes from five liver donors were treated with PB or CITCO as detailed in Materials and Methods. As expected, both PB and CITCO markedly induced the expression of CYP2B6 (Figure 2A), the prototypical human CAR target gene; however, only PB not CITCO robustly increased the expression of SLC13A5 (Figure 2B). In separate experiments, HPH were treated with PB in the presence or absence of SPA70 (a human PXR deactivator) [33] or CINPA1 (a human CAR inhibitor) [34]. Notably, the PB-induced SLC13A5 mRNA expression was nearly eliminated by SPA70 at 2.5 µM in human hepatocytes from two liver donors (HPH #163 and #167) (Figure 2C). On the other hand, CINPA1 only modestly repressed PB-induced SLC13A5 by approximately 13–25% (Figure 2D).

Consistent with their selective inhibition effects on PXR and CAR, respectively, SPA70 abolished RIF-mediated induction of CYP3A4 (Figure 2E), a typical PXR target gene, and CINPA1 was able to markedly suppress CITCO-mediated induction of CYP2B6 by 60–70% (Figure 2F). In addition to pharmacological inhibition, lentivirus small hairpin RNA was used to knockdown the PXR expression in HPH. As shown in Figure 3, knockdown of PXR expression subsequently results in attenuation of PB-mediated induction of SLC13A5 and CYP3A4. Together, these results strongly support that PXR plays a pivotal role in PB-mediated SLC13A5 induction, while activation of CAR may not be associated with such induction.

### 3.3. PB Induces the SLC13A5 Expression in CAR-KO but Not PXR-KO HepaRG Cells

HepaRG cells have been characterized as a promising surrogate of HPH for in vitro metabolism and toxicology studies [35]. The WT, CAR-KO, and PXR-KO HepaRG cell lines obtained from Sigma-Aldrich provide unique research tools for studying CAR- and PXR-dependent gene transcription [36,37,38]. We first validated the CAR-KO and PXR-KO HepaRG cell lines by assessing the expression and transcriptional activity of CAR and PXR, respectively. As expected, genetic modifications led to the production of nonfunctional CAR and PXR proteins that were barely detectable using the monoclonal antibody of CAR or PXR, individually (Figure 4A). Notably, CITCO and RIF robustly induced the expression of CYP2B6 and CYP3A4 in WT HepaRG cells, while such inductions were fully abolished in CAR-KO and PXR-KO cell lines, respectively (Figure 4B). Our subsequent data showed that PB-induced expression of SLC13A5 in HepaRG cells was markedly attenuated in the PXR-KO line, while its induction of CYP3A4 was partially affected (Figure 4C). Interestingly, PB exhibits enhanced induction of SLC13A5 in CAR-KO cells in comparison with WT HepaRG cells; its induction of CYP2B6 as expected was significantly repressed (Figure 4D). Collectively, these observations provide strong evidence suggesting that PB induces SLC13A5 expression in a CAR-independent manner that is most likely via the activation of PXR.

### 3.4. Identification of Novel PXR-Response Elements in the SLC13A5 Gene

To delineate the molecular mechanism(s) underlying PXR-regulated *SLC13A5* gene expression, we conducted in silico analysis of the *SLC13A5* gene sequence spanning −30 kb upstream of the transcription start site (TSS) to +1.5 kb downstream of the stop codon using the NUBIScan version 2.0 (http://www.nubiscan.unibas.ch/; accessed 6 July 2019). Nine potential PXR binding motifs with raw scores of 0.72 and above were identified at both the upstream and intron regions of SLC13A5 (Figure 5A), including the DR4-1, a previously validated distal PXR binding site [39]. Utilizing in vitro EMSA, we found that heterodimer of PXR and RXR binds strongly to DR4-1, DR4-I1, and DR4-I2 (Figure 5B), located in −22 kb, intron 6, and intron 7, respectively. To assess the functional relevance of the PXR-binding elements in SLC13A5 transactivation, an SLC13A5 luciferase reporter construct containing these three binding sites as depicted in Figure 5C, was cotransfected with PXR or CAR1 + A in HepG2 cells. Treatment with PB significantly enhanced the SLC13A5 reporter activity via PXR activation (Figure 5D) but not CAR (Figure 5E). CITCO and RIF were used as selective activators of CAR and PXR, respectively. In a separate experiment, HPH were transfected with the SLC13A5 luciferase construct without exogenous PXR or CAR. Similar to the results observed in HepG2 cells, PB and RIF but not CITCO efficiently enhanced the luciferase activity of SLC13A5, most likely through activation of the endogenous PXR expressed in HPH (Figure 5F). These data emphasize that PB-mediated induction of SLC13A5 transcription in the liver is a PXR-dependent event and several enhancer models contribute significantly to this induction.

### 3.5. Metformin Represses PB-Induced SLC13A5 Expression through PXR Inhibition

Prior studies indicated that metformin dramatically suppresses PXR activity in human hepatocytes [40]. To test whether metformin as a metabolic modulator can repress PB-mediated SLC13A5 expression, we first validated the suppressive effects of metformin on PXR-mediated transactivation of its prototypical target gene, CYP3A4, in HPH culture. As expected, metformin significantly suppressed both PB- and RIF-induced expressions of CYP3A4 in a concentration-dependent manner (Figure 6A). The subsequent study demonstrated that PB-induced SLC13A5 induction was robustly repressed by metformin at both mRNA and protein levels (Figure 6B,C). Moreover, metformin at 1 mM fully abolished PB stimulated activation of the SLC13A5-1k+DR4-1/I1/I2 or SLC13A5-(DR4-1)_3_ luciferase reporter activity when transfected in HPH (Figure 6D,E). Together, these results further support PB induction of SLC13A5 expression through PXR activation.

## 4. Discussion

PB, a first-line antiepileptic drug, is clinically used to treat and prevent the symptoms of seizures, sedation, hypnotics, insomnia, and infant jaundice. It has also been extensively used as a model compound eliciting hepatic induction of numerous drug-metabolizing enzymes and transporters. Our study identified that PB is a potent inducer of SLC13A5 in cultured human hepatocytes through activation of the PXR signaling pathway. Using pharmacological inhibition and genetic modification, we showed that disruption of PXR signaling dramatically suppresses PB-induced expression of SLC13A5, while inhibition of CAR does not affect this induction, even though PB was regarded as a prototypical CAR activator. Moreover, we identified two new DR4 response elements located in intron 6 and intron 7 of SLC13A5, along with the known distal upstream DR4-1, displaying a strong binding capacity to the PXR/RXR heterodimer. Our data also demonstrate that PB efficiently activates the luciferase activity of an SLC13A5 construct containing these three enhancer modules via PXR activation in HepG2 and HPH, while PB-mediated SLC13A5 transactivation was robustly suppressed by metformin as a PXR deactivator (Figure 7).

CAR and PXR are two closely linked and liver-enriched nuclear receptors that regulate the transcription of numerous drug-metabolizing enzymes, uptake and efflux transporters, and genes associated with energy homeostasis [28,41,42]. The inductive feature of PB in the liver has long been considered a consequence of CAR activation. Indeed, in mice, the knockout of CAR eradicates the highly pleiotropic effects of PB including CYP induction, alteration of energy metabolism, and promotion of liver tumor development [43,44]. Recent studies reveal that PB, while a mouse CAR but not a PXR activator, activates both human CAR and PXR [27,45,46]. In the current study, PB markedly induced the expression of SLC13A5 in human hepatocytes to the similar extent resulted by RIF, a typical human PXR activator. In contrast, treatment with CITCO, a human CAR agonist, only marginally affected SLC13A5 expression, while both PB and CITCO robustly increased the expression of CYP2B6 as a classic CAR target gene. Moreover, consistent with its species-selective feature of PXR activation, PB treatment did not induce hepatic expression of mouse Slc13a5 (Appendix A), which was upregulated by pregnenolone 16α-carbonitrile, a prototypical mouse PXR activator, whereas to a lesser extent [47]. Together, these findings suggest that SLC13A5 may not be a shared target for both CAR and PXR. It is well known that CAR and PXR share many common ligands and target genes, making delineation of the precise role of each individual receptor challenging. Recently, Taosheng Chen and colleagues identified CINPA1 and SPA70 as selective and potent inhibitors of human CAR and PXR, respectively [33,34]. Utilizing these chemical tools, we found that cotreatment with SPA70 almost entirely abolished PB-induced SLC13A5 expression, while CINPA1 only moderately affected this induction, further supporting the predominant role of PXR in PB-mediated induction of SLC13A5. It is worth noting, however, that while named “CAR inhibitor not PXR activator 1,” CINPA1 can also bind to and activate PXR [48]; thus, related data need to be cautiously interpreted.

The HepaRG cells have been established as a promising alternative to HPH, exhibiting hepatocyte-like morphology and functions, and expressing key liver-enriched transcription factors including CAR and PXR [37,38]. Both CAR-KO and PXR-KO HepaRG cell lines generated using the Zinc Finger Nuclease technology were obtained from Sigma-Aldrich and functionally validated in this study. We found that PB-induced SLC13A5 expression in WT HepaRG cells was completely abolished in PXR-KO but not in CAR-KO cells. It is noteworthy that disruption of CAR appears to enhance PB induction of SLC13A5 in comparison to WT HepaRG cells, a phenomenon that might be associated with the removal of CAR from competing with PXR in binding to the DR4 enhancer elements. Similar interplay between CAR and PXR that affects each other’s activities was documented previously [49,50].

We previously identified and functionally characterized an enhancer model (DR4-1) located at distal upstream (−22 kb) of the *SLC13A5* gene TSS that is associated with regulation of PXR-mediated SLC13A5 induction [19]. Further computational analysis of the *SLC13A5* gene spanning −30 kb upstream of TSS to downstream +1.5 kb from the stop codon revealed nine potential PXR response elements. Two DR4 from intron 6 and intron 7 as well as the DR4-1 demonstrated strong binding to the PXR/RXR heterodimer and were selected for the subsequent functional evaluation. PB and RIF but not CITCO significantly increased the luciferase activity of the SLC13A5-1k+DR1-1/I1/I2 construct when PXR was cotransfected in HepG2 cells, which is in line with recent findings that PB can efficiently bind to and activate human PXR [27]. In contrast, none of the compounds enhanced the luciferase activity of SLC13A5 with the cotransfection of CAR1 + A, a chemical responsive human CAR expression plasmid [30]. Unlike immortalized hepatoma cells, HPH express physiologically relevant levels of both CAR and PXR. Our data showed that PB-stimulated activation of the endogenous PXR in HPH was able to provoke SLC13A5 transactivation. Although these findings clearly indicate PXR but not CAR is responsible for PB-mediated induction of SLC13A5, we observed an obvious discrepancy between the robust induction of SLC13A5 mRNA expression and the relatively moderate activation of the luciferase activity, which hints at the potential existence of additional yet unidentified enhancer modules.

Elevated expression of SLC13A5 has been observed from liver samples of obesity, type 2 diabetes, and non-alcoholic fatty liver disease [13]. The subsequently increased intracellular citrate enhances de novo lipogenesis and gluconeogenesis in hepatocytes, contributing to the development of metabolic disorders. Metformin, a biguanide agent, is currently the most commonly used antidiabetic drug for the treatment of type 2 diabetes. Previous studies showed that metformin is also an efficient deactivator of both CAR and PXR [40,51] and can repress PXR expression [52]. Most recently, Kopel et al. reported that metformin inhibits the expression of SLC13A5 in HepG2 cells while at relatively high concentrations (5–10 mM) [53]. In the current study, we found that metformin markedly suppressed PB-induced SLC13A5 expression in a concentration-dependent manner (0.1–1 mM). In HPH, metformin at 1 mM completely abolished the increased mRNA expression and luciferase activity of SLC13A5 after PB treatment. While these data support SLC13A5 as another factor that attributes to metformin’s therapeutic effect on diabetes, alteration of SLC13A5 in the liver versus in the central nervous system may elicit opposite clinical benefits. Indeed, a recent case report suggests administration of metformin in a patient with SLC13A5-associated early infantile epileptic encephalopathy-25 (EIEE-25) may increase the psychiatric episodes, while discontinuation of metformin revealed improved clinical manifestation [54]. It is important to point out that PB has long been used for epilepsy treatment. A recent report indicates that PB appears to be one of the most effective drugs in SLC13A5-associated neonatal epilepsy [55]. Therefore, assuming PB could induce SLC13A5 expression in the neuron, it may provide a rationale for PXR activators as a potential new treatment for EIEE-25.

In conclusion, we uncovered that PB is a potent inducer of the *SLC13A5* gene in the human liver through a CAR-independent pathway by activation of PXR. Pharmacological and genetic inhibition of PXR but not CAR disrupts PB-mediated induction of SLC13A5 in both HPH and HepaRG cells. We identified two new intron DR4 enhancers that together with the distal DR4-1 contribute to PB induction of SLC13A5 via PXR-mediated transactivation. However, a large discrepancy between the mRNA induction and activation of the SLC13A5 reporter construct still exists, suggesting we have not yet identified all functional PXR response elements. Moreover, metformin markedly represses PB-induced SLC13A5 expression through the inhibition of PXR activity. Whether PB could induce neuronal SLC13A5 expression and through which ameliorate EIEE-25 symptoms warrants future investigations.

## Figures and Tables

**Figure 1 cells-10-03381-f001:**
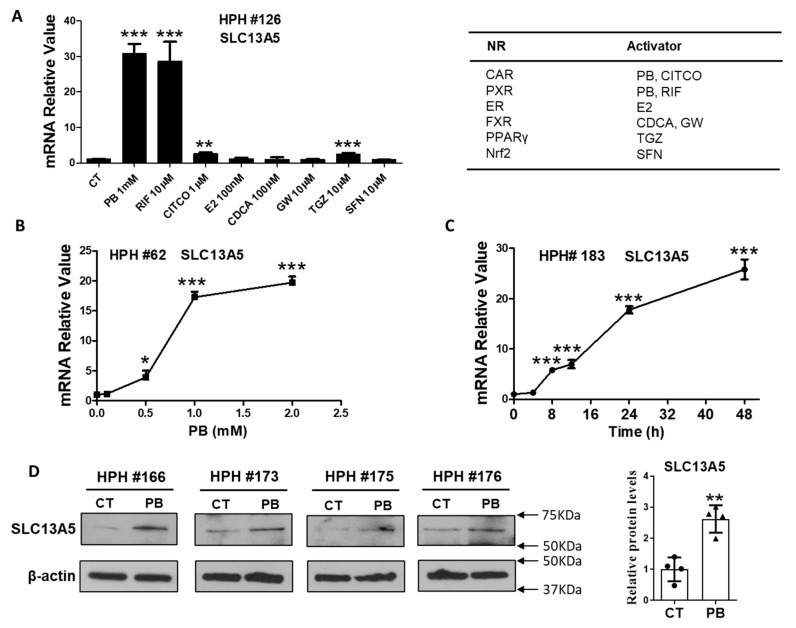
PB induces SLC13A5 expressions in HPH. Human hepatocytes prepared from liver donors (HPH #126) were treated with multiple nuclear receptor activators for 24 h as indicated. The expression of SLC13A5 mRNA was measured using RT-PCR assays (**A**). Concentration- (**B**) and time-dependent (**C**) induction of SLC13A5 mRNA by PB was detected using RT-PCR in HPH from liver donors #62 and #183, respectively. In separate experiments, HPH from liver donors (#166, #173, #175, and #176) were treated with vehicle control (CT) or PB (1 mM) for 48 h to analyze proteins by western blot assay (**D**). Data are expressed as mean ± S.D. (*n* = 3–4), * *p* < 0.05; ** *p* < 0.01; *** *p* < 0.001.

**Figure 2 cells-10-03381-f002:**
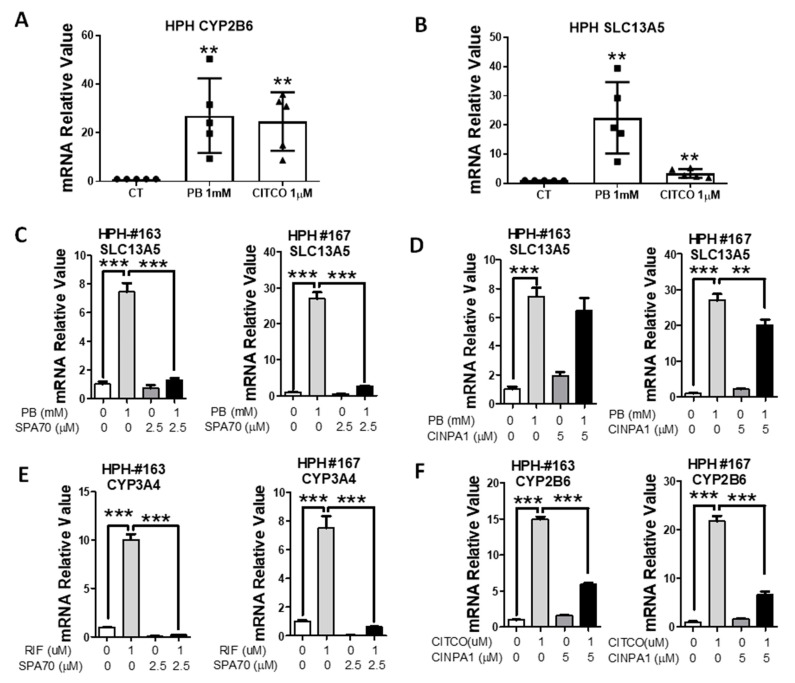
PXR inhibitor suppresses PB-mediated induction of SLC13A5 in HPH. HPH prepared from five liver donors were treated with 0.1% DMSO (CT), PB (1 mM), or CITCO (1 μM) for 24 h. RT-PCR analysis was used to measure the expression of CYP2B6 (**A**) and SLC13A5 (**B**). In separate experiments, HPH from liver donors (HPH #163, #167) were treated with 0.1% DMSO, PB (1 mM), RIF (10 µM), CITCO (1 µM), or in combination with SPA70 (2.5 µM) (**C**,**E**), and CINPA1 (5 µM) (**D**,**F**) as detailed in Materials and Methods. RT-PCR was used to measure mRNA expression of SLC13A5 (**C**,**D**), CYP3A4 (**E**), and CYP2B6 (**F**). Data are expressed as the mean ± S.D. (*n* = 3−5), ** *p* < 0.01; *** *p* < 0.001.

**Figure 3 cells-10-03381-f003:**
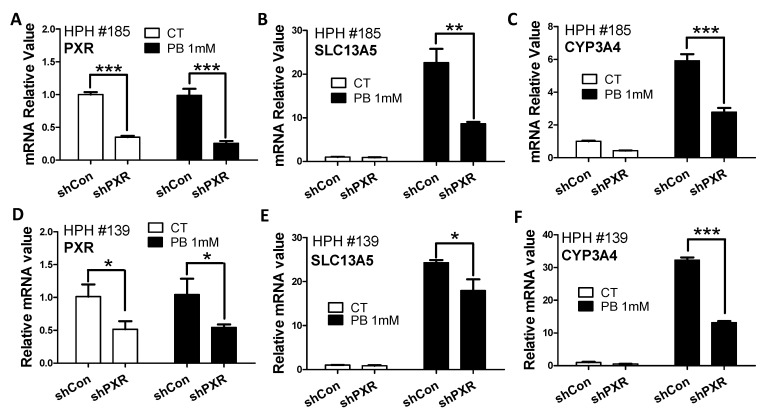
Knockdown of PXR affects PB-induced SLC13A5 expression in HPH. Human hepatocytes from liver donors (HPH#139, #185) were infected with lentiviral/PXR/small hairpin RNA (shPXR) or lentiviral-negative control (shCon) followed by treatment with vehicle control (CT) or 1 mM PB as detailed in Materials and Methods. Expressions of PXR (**A**,**D**), SLC13A5 (**B**,**E**), and CYP3A4 (**C**,**F**) were analyzed by RT-PCR. Data are expressed as mean ± S.D. (*n* = 3), (* *p* < 0.05; ** *p* < 0.01; *** *p* < 0.001).

**Figure 4 cells-10-03381-f004:**
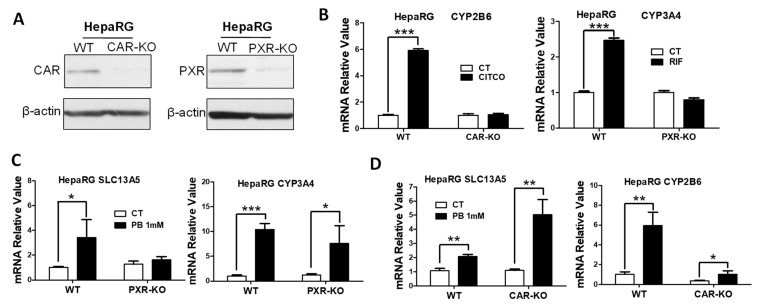
PB induces SLC13A5 expression in HepaRG cells independent of CAR. (**A**) After 21 days of differentiation, cell homogenates prepared from WT, CAR-KO, and PXR-KO HepaRG cells were subjected to western blotting analysis. (**B**) Differentiated WT, CAR-KO, or PXR-KO HepaRG cells were treated with vehicle control (CT), 1 µM CITCO, or 10 µM RIF for 24 h before RT-PCR analysis of CYP2B6 and CYP3A4 expression. After 72 h treatment with CT or PB (1 mM), the mRNA expression of SLC13A5 and CYP2B6 (**C**), or SLC13A5 and CYP3A4 (**D**), were measured in differentiated WT-, PXR-KO-, and CAR-KO-HepaRG cells. Data represent the mean ± S.D. (*n* = 3), * *p* < 0.05; ** *p* < 0.01; *** *p* < 0.001.

**Figure 5 cells-10-03381-f005:**
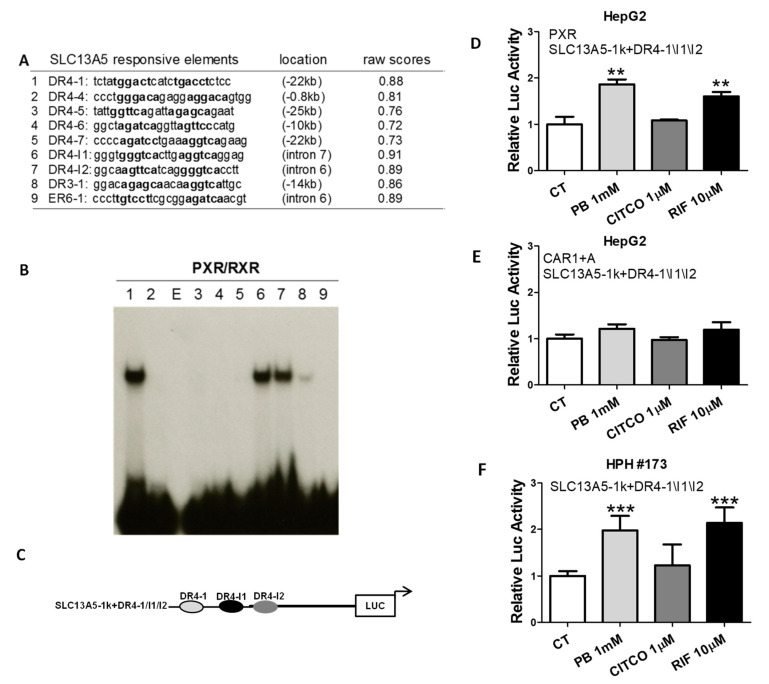
Identification of PXR-response elements in the upstream and introns of *SLC13A5* gene. (**A**) Nine PXR-response elements were identified in the upstream and introns of *SLC13A5* gene. (**B**) In vitro binding of the predicted DR4, DR3, and ER6 elements with PXR/RXRα heterodimer was measured using electrophoretic mobility shift assays. Lanes 1–9 represent elements summarized in Table A; E: empty lane. (**C**) Schematic illustration of the SLC13A5 reporter construct, namely, SLC13A5-1k+DR4-1\I1\I2. HepG2 cells were cotransfected with SLC13A5-1k+DR4-1\I1\I2 in the presence of human PXR or CAR1 + A expression vector for 24 h (**D**,**E**). In a separate experiment, SLC13A5 1k+DR4-1\I1\I2 was transfected in HPH #173 (**F**). All transfected cells were treated with 0.1% DMSO (CT), PB (1 mM), CITCO (1 µM), or RIF (10 µM) for 24 h. Luciferase activities were measured using the Dual-Luciferase reagent (Promega) in cell extracts according to the manufacturer’s instructions. Data represent the mean ± S.D. (*n* = 3), ** *p* < 0.01; *** *p* < 0.001.

**Figure 6 cells-10-03381-f006:**
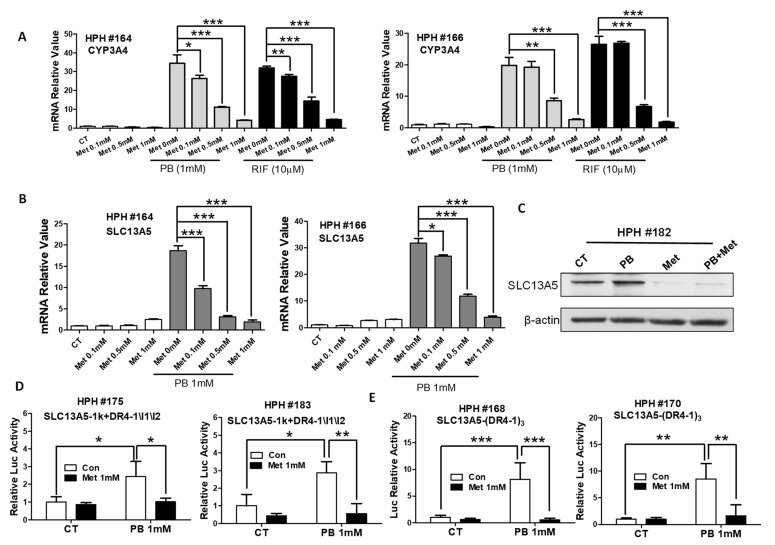
Metformin represses PB-mediated SLC13A5 expression by inhibition of PXR activation. HPH were treated with vehicle control (CT), PB (1 mM), Met (0.1, 0.5, 1 mM), or their combinations as indicated for 24 h. Total RNA extracted from hepatocytes was subjected to RT-PCR analysis of CYP3A4 and SLC13A5 expression (**A**,**B**). Homogenate proteins from CT, PB (1 mM), Met (1 mM), or PB + Met treatments were subjected to immunoblotting analysis of SLC13A5 and β-actin expression (**C**). HPH transfected with the SLC13A5-1k+DR4-1\I1\I2 (**D**) or SLC13A5-(DR4-1)3 (**E**) reporter construct as described in Materials and Methods were treated with CT, PB (1 mM) with and without Met (1 mM) for 24 h before the determination of luciferase activities. Data represent the mean ± S.D. (*n* = 3), * *p*< 0.05; ** *p* < 0.01; *** *p* < 0.001.

**Figure 7 cells-10-03381-f007:**
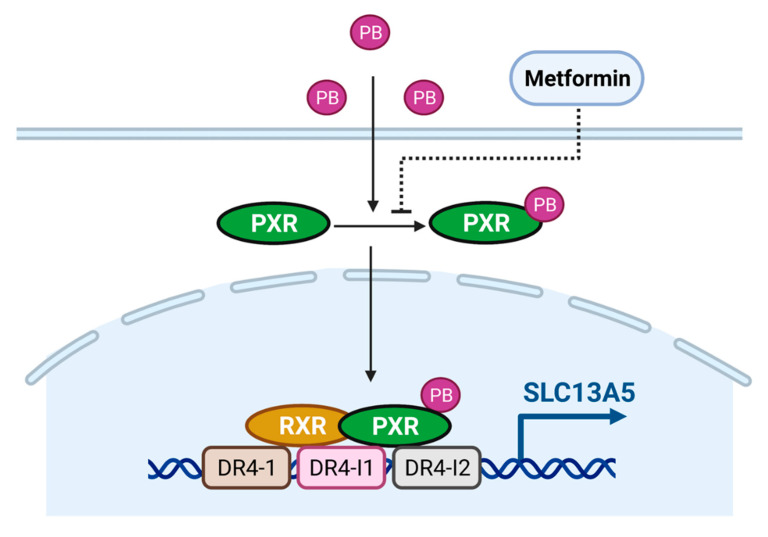
Schematic illustration of PB as an inducer of SLC13A5 through activation of human PXR. PB is a ligand of human PXR, which translocates PXR from the cytosol to the nucleus of human hepatocytes. Once inside the nucleus, PXR and RXR form a heterodimer that binds to the upstream DR4-1, intron DR4-I1, and DR4-I2, and stimulates the *SLC13A5* gene transcription.

## Data Availability

All data are included in figures in the manuscript and as Appendix A.

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
