# Peer review of "Phenobarbital Induces SLC13A5 Expression through Activation of PXR but Not CAR in Human Primary Hepatocytes"

_cells, 2021, doi:10.3390/cells10123381_

Round 1
Reviewer 1 Report
Review
In the manuscript, Li et al. proposed phenobarbital (PB) as an inductor of the SLC13A5 gene through a PXR dependent mechanism. By using primary human hepatocytes and HepaRG cells, they describe SLC13A5 induction by PB and rifampicin. Using pharmacological inhibition and genetic ablation, the authors describe the induction as PXR-mediated. Furthermore, they identified tree PXR enhancer modules located in promotor and in introns of the SLC13A5 gene and functionally described them using electrophoretic mobility shift assays and luciferase reporter experiments. Finally, they identified SLC13A5 downregulation mediated by metformin after PXR activation.
The study contains rigorous mechanistic data on PXR-mediated regulation of the SLC13A4 gene and on PXR binding to SLC13A5 enhancer modules. Authors also convincingly document that metformin suppresses this regulation.
In the Discussion section, the authors discuss a link with infantile epileptic encephalopathy suggesting the clinical relevance of this study. However, because SLC13A5 is predominantly expressed in the liver, I lack discussion about the possible effects of SLC13A5 induction on liver metabolism. In summary, the authors provided solid and convincing data on PXR-specific SLC13A5 induction with PB in human hepatocyte models.
Minor comments:
- In the abstract authors used the term „metabolic syndromes“. Metabolic syndrome is well-defined pathology spelled in the singular.
- Data in Figure 2D should be discussed in more detail. The paper by Jaske et al. Arch Toxicol
2017 Jun;91(6):2375-2390 should be considered in the discussion.
3. Crosstalk of PXR and CAR may be considered in the regulation of SLC13A5 in the manuscript.
Author Response
Thank you for comments and constructive suggestions. Detailed responses are included in the attached files.

Reviewer 2 Report
With the manuscript “Phenobarbital induces SLC13A5 expression through activation of PXR but not CAR in human primary hepatocytes”, authors reported the role of PXR in phenobarbital-mediated induction of SLC13A5 gene in human. The results presented in this paper are novel and interested by researchers in the field of drug metabolism and toxicology, since phenobarbital is well known as CAR activating drug and the gene induction by phenobarbital is generally considered through CAR activation. PXR and CAR have numerous common target genes, and therefore, we cannot screen the ligand for PXR by the induction of its target gene such as CYP3A4 (because CAR activation also induces the CYP3A4 mRNA levels). The results presented in this paper indicate the possibility of SLC13A5 gene as the typical evaluation indicator of PXR activation. Data is clear and convincing. To ensure the data interpretation, I have a few suggestions and questions should be considered by authors.
- Since phenobarbital is considered as CAR specific activator in rodents, species difference of the regulation of SLC13A5 gene induction should be discussed.
- Authors used metformin as a PXR in Fig6, although they used representative PXR inhibitor SPA70 in Fig2. Reviewer understood that authors recently found that metformin suppressed PB-induced SLC13A5 expression. But reviewer couldn’t understand the author’s intension to use these two inhibitors and showed in different figures, Fig2 and Fig6. What is the difference between these two inhibitors?
- What does the squared PXR means in Fig7? The legend for Fig7 requires more explanation. The figure legend should provide enough information for the reader to understand its significance in the absence of referring back to the text.
Minor
- In the reporter assay in HepG2 cells, phenobarbital increased PXR dependent transcription. Reviewer think this is important information which indicating that phenobarbital is direct ligand for PXR, while phenobarbital is indirect activator of CAR. It better to be described in the manuscript.
Author Response
Thank you for the comments and suggestions. Detailed responses have been included in the attached files.

Reviewer 3 Report
In the manuscript “Phenobarbital induces SCL13A5 expression through activation of PXR but not CAR in human primary hepatocyte” Li et al. showed that phenobarbital regulates human SLC13A5 in a PXR-dependent but CAR-independent manner. Pertinent cellular models including primary human hepatocytes, HepaRG and HepG2 were used. The comparison of SLC13A5 regulation by prototypical inducers to that of CYP3A4 and CYP2B6 allows to discriminate convincingly between CAR- and PXR-mediated regulation.
In a general point of view, the manuscript is clear and very well written. The experiments are well designed and conducted and the results are convincing.
Nevertheless, some points have to be clarified:
- Figure 1: Dexamethasone activates PXR at supra-micromolar concentrations and induces CYP3A4 (PMID: 11737189). No induction was observed for SLC13A5. Discuss this point.
- Indicate the size of SLC13A5 on western-blot (Figure 1D)
- Figure 3: Does shPXR affect CAR expression?
- Figure 5E: As CAR is constitutively active, its transfection is sufficient to induce the expression of target genes. Showing SCL15A3-1k+DR4-1/I1/I2 luciferase activity following the transfection of empty vector in comparison to the transfection of CAR1+A vector will be more informative (i.e. basal expression).
Add the name of the empty control vectors encoding CAR or PXR in M&M.
- Metformin suppresses PXR-mediated CYP3A4 transcriptional activity but it is also the case for other nuclear receptors, including CAR (PMID: 21920351). Moreover, metformin inhibits PXR expression (PMID: 28238946). This must be discussed.
- Discussion lines 370-372 “disruption of CAR appears to enhance PB induction …. “ . This balance between PXR and CAR activities in link with their level of expression was observed by others and may be discussed (PMID: 29269410; PMID: 15833898)
- Graphical abstract: Responsive elements can be added (like in figure 5C). Illustrate the fact that CAR is not involved in SLC13A5 regulation by PB.
Author Response
Thank you for review our manuscript. All comments are addressed in the attached files.
